# Targeted Therapeutic Strategies for the Treatment of Cancer

**DOI:** 10.3390/cancers16020461

**Published:** 2024-01-22

**Authors:** Benjamin Victoir, Cécile Croix, Fabrice Gouilleux, Gildas Prié

**Affiliations:** 1INSERM UMR 1100 CEPR, Equipe “Mécanismes Protéolytiques Dans L’inflammation”, Faculté de Médecine, 10 Boulevard Tonnellé, BP 3223, 37032 Tours Cedex 01, France; benjamin.victoir@etu.univ-tours.fr (B.V.); cecile.croix@univ-tours.fr (C.C.); gildas.prie@univ-tours.fr (G.P.); 2INSERM UMR 1100 CEPR, Equipe “Infection Respiratoire et Immunité”, Faculté de Médecine, 10 Boulevard Tonnellé, BP 3223, 37032 Tours Cedex 01, France

**Keywords:** targeted therapies, multitarget therapies, drug release, vectorization, drug combinations, nanocarriers

## Abstract

**Simple Summary:**

Today, a significant issue in treating the most common cancers is their resistance to current treatment options like chemotherapy, radiotherapy, or immunotherapy. Drug combinations and/or dual inhibitors offer a promising alternative strategy to overcome this resistance. The purpose of this review is to present various existing options for the simultaneous administration of distinct drugs.

**Abstract:**

Extensive research is underway to develop new therapeutic strategies to counteract therapy resistance in cancers. This review presents various strategies to achieve this objective. First, we discuss different vectorization platforms capable of releasing drugs in cancer cells. Second, we delve into multitarget therapies using drug combinations and dual anticancer agents. This section will describe examples of multitarget therapies that have been used to treat solid tumors.

## 1. Introduction

In 2020, cancer caused almost 10 million deaths worldwide, with slightly over 19 million new reported cases. The four most common types of cancer are breast, lung, colorectal, and prostate cancer. While many treatment options exist to fight cancer, patients are increasingly experiencing resistance effects and relapse. This major problem is caused by the heterogeneity of tumors, which can be inter- or intratumoral. Intertumoral heterogeneity refers to differences between patients, which can be caused by individual factors such as germline genetic variations or environmental factors. Intratumoral heterogeneity, on the other hand, refers to tumoral variations within a single patient, either across tumoral sites or even within the same site [1]. These phenomena are explained by various mechanisms related to intrinsic or extrinsic causes. Some factors are related to genotypic changes, such as mutations, gene amplifications, and chromosomal rearrangements. Other factors are associated with the inactivation of anticancer drugs through adaptive mechanisms, such as the inhibition of cell death or even multidrug resistance mediated by higher drug efflux [2].

Laboratories around the world are actively researching new treatments to meet these challenges. One possible approach involves the precise targeting of cancer cells through targeted therapies. The goal is to deliver a drug to cancer cells in the form of a prodrug. This prodrug should remain inactive during its journey through the body and then be activated by a carefully chosen mechanism based on the target and the nanocarrier [3].

Additionally, the development of dual inhibitors that can target multiple proteins simultaneously is believed to have the potential to overcome cancer cells’ resistance to chemotherapy [4]. Another option within this strategy is to combine two drugs, but careful consideration should be given to minimize potential adverse effects. These combinations may have additive, synergistic, or antagonistic effects. Therefore, optimization is a crucial step when evaluating the combined use of different drugs.

The aim of this review is to first outline the different types of targeted therapies that have been effective in the past, along with their drug release mechanisms. We will then discuss the effectiveness of drug combinations and the efficiency of inhibiting multiple biological targets to combat cancer resistance.

## 2. Targeted Therapies

For decades, the primary objective in oncology has been the precise targeting of cancer cells. This strategy aims to minimize the severe side effects frequently experienced by patients and counter the increasingly observed resistance phenomena during the treatment process. Targeted therapy is conceived within this perspective, seeking to prevent undesirable effects and selectively target specific proteins to circumvent resistance phenomena. To develop cancer therapies, researchers have created nanocarriers, which are vectorization platforms made of organic molecules that transport drugs through encapsulation or conjugation. In the following sections, we will explore the various nanocarrier platforms developed in recent years and evaluate their effectiveness in enhancing cancer treatment. We will also examine the major drug release mechanisms.

### 2.1. Nanocarriers

#### 2.1.1. Liposomes

The liposome is the first identified carrier with the potential to serve as a platform for delivering drugs. Liposomes are spherical entities with an aqueous core, consisting of alternating lipid and aqueous layers made of amphiphilic compounds with lipidic chains and hydrophilic heads (Figure 1) [5]. They can vary in size from 20 nanometers to several micrometers. Liposomes are significant because they can encapsulate drugs for transport within the body. Liposomes have a notable advantage due to their envelope, which can be tailored with ligands to target specific release sites or enhance the stability and pharmacokinetics of the platform. For instance, the incorporation of single-stranded oligonucleotides or aptamers onto the surface of liposomes enables the specific targeting of small molecules, proteins, or even intact cells. Aptamers are selected using SELEX (Systematic Evolution of Ligands by Exponential Enrichment), a method based on the screening of a random nucleic acid library. Several Aptamers targeting cancer biomarkers have been successfully used in the treatment of solid and hematopoietic cancers. Liposomes are then important relevant drug carriers for targeting cancers [6]. 

To date, 14 liposomal carriers have been approved for clinical use. The first was pegylated liposomal doxoribucin Caelyx in 1995 for the treatment of ovarian cancer, followed by daunorubicin formulation DaunoXome in 1996 for Kaposi’s sarcoma. The most recent approval is for Arikayce, an amikacin liposomal inhalation suspension, which has been available in Europe since 2020 for the treatment of lung disease. Liposomal carriers can be administered by a variety of routes, including intravenous infusion and intramuscular and intrathecal injection, as well as epidural and local infiltration. Most are small unilamellar vesicles between 30 and 100 nanometers in size. Approved carriers consist mainly of glycerol phospholipids, sphingomyelin, and cholesterol. Liposomes are either charged or neutral at physiological pH, which gives them targeting capabilities or enables them to avoid aggregation in the presence of a strong Coulombic repulsive force, such as in the daunorubicin/cytarabine liposomal injection form Vyxeos [7]. 

Another advantage of the liposomal carrier is its ability to carry two or more drugs, providing an efficient solution to overcome chemoresistance. For example, in 2012, Isacchi et al. co-encapsulated both artemisinin and curcumin to treat drug-resistant malaria infection [8]. On the other hand, in 2020, Jin et al. developed a synergistic therapy to treat lung cancer by co-encapsulating betulinic acid, parthenolide, honokiol, and ginsenoside Rh2. This clearly shows that combining drugs in a liposomal system seems to be a relevant solution, bearing in mind that this quadruple therapy also shows fewer side effects than cisplatin monotherapy, which induces kidney damage [9].

#### 2.1.2. Polymeric Nanoparticles

Polymeric nanoparticles serve as robust and biocompatible colloidal carriers, often exhibiting biodegradability (Figure 2) [10]. They are highly versatile in nanomedicine applications due to their structural adaptability, which allows modulation of their properties. For example, their surfaces can be tailored to improve drug loading efficiency, biodistribution, pharmacokinetic control, and therapeutic efficacy [11]. 

In terms of composition, different types of polymers can be used, including synthetic polymers such as poly(lactic acid) and poly(ε-caprolactone), as well as natural polymers such as gelatin, dextran, and collagen. Depending on the polymers chosen, drugs are encapsulated by dispersion within the polymer matrix or by chemical conjugation, allowing for different release mechanisms [12]. For example, doxorubicin conjugated to dextran and subsequently encapsulated in a hydrogel showed reduced cytotoxicity and increased efficacy against cancer cells [13]. Another example is tamoxifen loaded into poly(lactic-co-glycolic acid) nanoparticles, which showed superior in vitro anticancer activity compared with the drug alone [14]. It is also possible to load two different drugs onto this type of nanoparticle, as demonstrated by Wang et al. in 2010, who conjugated paclitaxel and encapsulated combretastatin A4. This conjugation showed favorable anti-cancer and anti-vascular activities both in vitro and in vivo [15]. Thus, by appropriately modulating the polymer, polymeric nanoparticles have the potential to offer targeted treatment options for various cancers.

#### 2.1.3. Micelles

Formed by the self-assembly of selected amphiphilic molecules, micelles consist of a hydrophobic core and a hydrophilic shell (Figure 3). Micelles form spontaneously under certain conditions, depending on the critical micelle concentration (CMC) and temperature. Organic drugs can be encapsulated in the micelle core, making them water soluble and transportable in the bloodstream. Incorporation into micelles occurs through physical, chemical, or electrostatic interactions [12]. For example, doxorubicin encapsulated in a micelle based on cationic 1,2-dioleoyl-3-trimethylammonium propane/methoxy poly(ethylene glycol) (DPP) showed effective anticancer effects against bladder cancer [16]. Similarly, in 2011, Shin et al. obtained significant results regarding the solubilization and toxicity of three different drugs incorporated into a polymeric micelle, revealing the potential strategy of multiple drug release using this type of nanocarrier [17].

#### 2.1.4. Dendrimers

Dendrimers are organic molecules that self-assemble into tree-like structures with many branches to which drugs can be conjugated in varying proportions depending on the dendrimer size (number of branches). The dendritic structure offers numerous advantages and applications, two of which are illustrated in Figure 4 [18]. One advantage is the ability to incorporate specific biological recognition blocks (Figure 4a), making them relevant for use as targeting nanocarriers [19]. In 2009, Tekade et al. synthesized a polyamidoamine-based dendrimer carrying both methotrexate and all-trans retinoic acid (ATRA) (Figure 4b). This dendrimer showed reduced hemolytic toxicity and enhanced cytotoxicity against HeLa cells compared with the individual drugs [20]. This example highlights the relevance of this type of nanocarrier for oncology applications. 

#### 2.1.5. Carbon Nanotubes

The application of carbon nanotubes in medicine emerged after 1985 following the discovery of fullerenes, the third allotrope of carbon. In 1991, Iijima et al. identified a new structural form of this allotrope called carbon nanotubes (CNTs). These tubes consist of layers of graphene rolled into cylinders with diameters on the order of nanometers and lengths that can extend to several micrometers. There are two types of CNTs, as shown in Figure 5: single-walled carbon nanotubes (SWCNTs) and multi-walled carbon nanotubes (MWCNTs) [21]. This type of carrier offers numerous advantages that are highly sought after for drug delivery, such as its strength, high drug loading capacity, and biocompatibility [22].

Despite limited biodegradability, with lifetimes ranging from months to years, and a certain toxicity that can be mitigated by functionalization, carbon nanotubes represent a promising avenue for drug vectorization and release. Some hypotheses suggest that their needle-like shape may be advantageous for penetrating the cell membrane [23]. 

These nanocarriers have shown favorable results in specific cancers such as brain, blood, and breast cancer [22]. For example, in the context of acute lymphoblastic leukemia, a system incorporating daunorubicin, an Sgc8c aptamer, and a single-wall carbon nanotube (SWCNT) showed enhanced internalization and selectivity toward the targeted Molt-4 cell line compared with a non-targeted U266 cell line, in contrast to daunorubicin alone [24]. 

#### 2.1.6. Antibody–Drug Conjugates

An ADC, or antibody–drug conjugate, is a very popular vectorization, targeting, and delivery system that has gained considerable attention over the past two decades. It consists of three essential components: an antibody, a cytotoxic agent, and a chemical linker (Figure 6). This ADC should remain stable while circulating in the bloodstream, exhibit precise targeting of the desired biological target, and ensure proper release of the cytotoxic agent. This highlights the importance of carefully selecting the antibody that specifically binds to the antigen located on the surface of the targeted cancer cell, selecting a suitable chemical linker that can potentially be cleaved under the conditions present in the tumor cell if required, and also selecting a cytotoxic agent capable of inducing cell death [25]. 

In 2000, the first antibody–drug conjugate (ADC) to be approved by the US Food and Drug Administration (FDA) was Mylotarg (gemtuzumab ozogamicin) for the treatment of acute myeloid leukemia [26]. This marked the beginning of a remarkable surge in ADC development. Since then, the ADC field has grown exponentially, with no fewer than 14 ADCs currently approved and over 100 in clinical trials [25]. 

Similarly, other types of bioactive molecules with anticancer properties, such as peptide nucleic acid (PNA), antisense oligonucleotides (ASO), photosensitizers, or siRNA, can be linked to antibodies [27,28,29,30]. For instance, siRNA can suppress oncogene expression by cleaving the messenger RNA that encodes it. In 2015, Bäumer et al. demonstrated the efficacy of an anti-EGFR antibody conjugated with a siRNA complex against KRAS for the treatment of colon cancer [30].

Another option is to conjugate photosensitizers, such as porphyrin, which induce the formation of singlet oxygen or ROS after photo-stimulation, acting as cytotoxic agents. This was demonstrated by Abu-Yousif et al., who conjugated a benzoporphyrin derivative to an EGFR-targeting antibody (cetuximab) to combat EGFR-positive ovarian cancer cells [29].

In addition, smaller antibody formats can be used to create fragment–drug conjugates (FDCs) by attaching cytotoxic agents to them. This can impact the half-life of FDC, tumor penetration, and tumor exposure, as demonstrated by Deonarain et al. [31]. 

There are various vectorization platforms available, offering a wide range of adaptable applications depending on the type of cancer being targeted. Ikwuagwu et al. and Zhu et al. have also described other carriers, such as virus-like drug conjugates and aptamer–drug conjugates [32,33]. In the following sections, we will focus on the most well-known release mechanisms that achieve the cytotoxic effect of the drug at the tumor site.

### 2.2. Drug Release Mechanisms

The goal of controlled release using nanocarriers is to maintain a relatively constant amount of drug at the desired site within the therapeutic window, the range between the minimum effective concentration (MEC) and the minimum toxic concentration (MTC). This release process is influenced by several factors such as the composition of the nanocarrier (drug, polymer, etc.), the drug/nanocarrier ratio, and physical and/or chemical interactions between the drug and the nanocarrier, among others [34]. Four main controlled release mechanisms are presented below: diffusion-based release, swelling-based release, external stimuli-based release, and chemical cleavage-based release.

First, diffusion release is a mechanism that occurs in systems called “capsules”, where the drug is dissolved or dispersed in the core. This core is surrounded by a membrane that may or may not have pores and may or may not be partially water soluble. The drug dissolves in the core and may cross the membrane by flux exchange (Figure 7) [35]. Diffusion is based on the principle that the concentration of a substance between two merging environments tends to become equal. Consequently, this phenomenon causes the drug to diffuse from the inside to the outside of the nanocarrier. As the drug concentration outside the capsule decreases due to its consumption by the protein, the concentration equilibrates, resulting in a continuous release over time. Diffusion can also occur within polymeric matrices where the drug is dispersed alone but without a membrane and will be rapid and decrease over time [34].

Second, drug release can be observed through the swelling of a polymer. This type of polymer must have a structure consisting of meshes of different sizes. Drugs dispersed in this network of meshes will be retained if they are at least one mesh in size. Under different conditions, this network will swell based on the balance between forces that limit deformation and osmosis that leads to water absorption, thus allowing drug release (Figure 8). Mesh swelling can vary based on several factors including the pH, temperature, and ionic strength of the environment [36]. For example, Carbinatto et al. prepared swellable matrix tablets of high amylose and pectin cross-linked with sodium trimetaphosphate, which showed high swelling ability. This system represents a promising carrier for this type of controlled drug delivery system [37]. 

Third, nanocarriers can respond to external stimuli, which can be applied to achieve significant spatio-temporal control in the context of targeted therapy. Indeed, outside the body, clinicians can non-invasively apply the desired stimuli wherever and whenever they wish. This technique ensures the ability to control delivery as internal stimuli can vary depending on the disease and/or patient. These external stimuli include light, temperature, ultrasound, magnetic fields, and even electric fields. For example, in the context of light activation, infrared radiation can trigger the destabilization or cleavage of photosensitive linkers, thereby enabling drug release. The advantage of this infrared radiation is its reduced attenuation by blood and soft tissues, allowing better tissue penetration to reach the target [38]. Another widely used and promising release mechanism is chemical cleavage. The following are different chemical drug releases that can occur depending on the trigger used within the nanocarrier. 

Acid-sensitive release: First, pH-sensitive mechanisms are particularly sensitive to the acidic conditions often found in cancer cells as opposed to healthy tissues. This mechanism can occur through the appearance of charge and/or hydrophilicity following a decrease in pH, leading to structural changes involving rearrangements, swelling, or even disassembly. Another observation is the loss of hydrophilicity with pH reduction, which destabilizes the endosomal membrane, resulting in drug release into the cell cytoplasm. Both of these scenarios are common in the administration of polymeric nanocarriers [39].

In addition, some selected covalent chemical bonds may be stable at neutral pH but sensitive to acidic conditions. Several chemical functionalities can serve this purpose, such as hydrazone, silyl ether, imine, cis-aconityl within the maleic amide family, acetal, ortho ester, or even β-thiopropionate [39]. For example, Wang et al. recently designed an efficient silyl ether-based acid-cleavable linker with improved stability compared with other pH-sensitive linkers. They conjugated monomethylauristatin E (MMAE) to this linker and observed effective payload release at acidic pH after 7 days at 37 °C, with 50% release at pH 5.5 and 100% release at pH 4.5. These values exceeded the 30% release at neutral pH under the same conditions (Figure 9) [40]. 

Redox-sensitive release: The tripeptide glutathione (GSH) is overexpressed in the tumor microenvironment of many cancers, reaching concentrations of 2–10 mM compared with 2–20 µM in healthy cells. The use of redox-sensitive linkers, such as those carrying a disulfide bond, is relevant in this reducing environment. It is noteworthy that the difference in GSH concentration increases the selectivity of targeted therapies using this type of redox-sensitive release [41]. An example of the release of the pyrrolobenzodiazepine dimer SG2057 using disulfide reduction is shown in Figure 10, demonstrating the cleavage of the ADC mAb-SG3231 for R = H and mAb-SG3451 for R = Me [42].

Enzyme-sensitive release: Enzyme-sensitive triggers are widely used in the design of cleavable linkers, especially in ADCs. The advantage of these triggers lies in the specific recognition of their known motifs by the respective enzymes. In addition, these enzymes are often overexpressed in tumor cells, making them excellent candidates for selective targeting of cancer cells. A classic example is the chemical linker recognized by cathepsin B, which carries a motif identified by the enzyme, consisting of known dipeptides [43] followed by the self-immolating 4-aminobenzylic alcohol spacer [44]. This recognition triggers cleavage and initiates the electron cascade leading to drug release (Figure 11).

Cleavage by β-glucuronidase is another well-established method based on the recognition of a “carbohydrate” motif. Once recognition occurs, the same type of electron cascade mentioned above is initiated, allowing drug release (Figure 12) [44].

## 3. Multi-Target Therapies

### 3.1. Generalities

As mentioned above, the resistance phenomena encountered in cancer treatment, together with the cytotoxic effects on healthy cells, pose significant challenges to researchers. Indeed, many known resistance mechanisms prevent patients from achieving complete remission. In addition, many undesirable side effects occur after treatment and reduce the quality of life of patients undergoing chemotherapy. To overcome these disadvantages, the scientific community is considering several solutions. First, the use of combination therapies could reduce doses and thus minimize side effects, while at the same time reducing the possibility of resistance. Second, the use of combinations of drugs may produce the same interesting results as combination therapies by targeting different disease-associated proteins. Finally, dual inhibitors targeting multiple pathways (kinases, growth factors, receptors, hormones, etc.) could reduce the potential side effects associated with the use of several different drugs. 

### 3.2. Combination Therapies

The most common combinations are chemotherapy with either radiotherapy or immunotherapy.

The combination of radiotherapy and chemotherapy has been shown to have a significant advantage over radiotherapy alone in fighting radioresistant cancers. The principle of radiotherapy is to deliver radiation to tumor cells as specifically as possible in order to destroy them. However, radiation monotherapy often leaves residual tumor cells that can lead to future relapse. In non-small cell lung cancer (NSCLC), factors such as tumor volume and hypoxia are correlated with radioresistance (Figure 13). However, increasing the dose of radiation to a given number of tumor cells until they are completely destroyed and the cancer is eradicated would affect healthy organs and tissues, making this technique unprofitable [45]. 

Therefore, the combination of both chemotherapy and radiotherapy could have an interesting effect on their respective doses. For example, the combination of gemcitabine and radiotherapy is relevant because gemcitabine, which in monotherapy is known to induce cell cycle arrest and apoptosis in human leukemia and solid cancer cells, acts as a radiosensitizer, making tumor cells more sensitive to radiation [46]. 

The principle of immunotherapy is to activate the immune system to target cancer or degenerative diseases. In the case of cancer, the progression of the disease leads tumor cells to adapt to their environment, contributing to the creation of an immunosuppressive microenvironment. This results in the loss of expression of certain tumor antigens, allowing cancer cells to evade immune surveillance [47]. Combining chemotherapy with immunotherapy can overcome this disadvantage. Indeed, tumor regression through the action of chemotherapy reduces the immunosuppressive nature of the microenvironment, which in turn allows immunotherapy to effectively activate the immune system [48].

There are several examples of such combination therapies, and the first major breakthrough occurred in the context of lung cancer, which has a PD-L1 expression level of more than 50%. The combination of pembrolizumab with a chemotherapy regimen consisting of pemetrexed and a platinum-based drug showed excellent efficacy compared with chemotherapy alone. The overall survival (OS) rate for chemotherapy alone was 49.4%, compared with 69.2% for the doublet. Progression-free survival (PFS) was 4.9 months and 8.8 months [49]. This dual therapy provided a significant survival benefit not only in lung cancer but also in several other cancers such as breast cancer and squamous cell carcinoma of the head and neck [50]. 

### 3.3. Drug Combinations

The combination of two chemotherapeutic agents is a method that has proven its effectiveness for many decades [51]. Around the middle of the 20th century, the first example of this type of treatment was for acute lymphoblastic leukemia, using a combination of antifolates, corticosteroids, and cytotoxic drugs. At the time, this was a remarkable achievement given that 100% of children with the disease would die [52]. Today, the goal is to continually discover drugs that have a synergistic or at least additive effect. This approach makes it possible to reduce doses in order to minimize side effects that would otherwise reduce patients’ quality of life. It also helps to overcome resistance mechanisms that have adapted to existing treatments for many types of cancer [53]. 

#### 3.3.1. Generalities

To evaluate drug combinations, it is necessary to observe the MI profiles of the drugs, which refers to the molecular interactions with individual biomolecules, pathways or processes related to pharmacodynamics, toxicology, pharmacokinetics, and combination effects. Once the MI profile has been assessed, the nature of the drug combination can be described. 

Pharmacodynamic drug–drug interaction studies have described different effects: (1) synergistic, where the therapeutic activity is greater than the sum of the effects of the individual drugs; (2) potentiative, where the activity of one drug is enhanced by another inactive drug; (3) additive, where the activity is the result of the sum of the effects; (4) antagonistic, where the activity is less than the sum of the effects; (5) reductive, where the activity of a single drug is reduced by another inactive drug; and (6) coalistic, where a single drug is inactive but is active in combination with another drug [54].

The most biologically interesting and sought-after interaction in drug combinations is synergy. This effect between two molecules can result from different actions of one molecule on the other. This synergy can be illustrated by the combination of celecoxib and emodin, which regulate the growth of certain cancer cells. Celecoxib suppresses cancer growth by inactivating the protein kinase AKT, while emodin inhibits the PI3K pathway, reducing the suppressive activity of AKT’ on apoptosis. This results in a synergistic effect between the two active molecules, making it a promising combination [55]. 

#### 3.3.2. Drug Combinations in Solid Tumors

Many extrinsic and intrinsic factors are involved in therapeutic resistance: the presence of cancer stem cells in tumors, changes in the tumor microenvironment, tumor immune evasion, overexpression of ABC or efflux transporters that reduce intracellular drug concentrations, oncogenic signaling, and metabolic alterations together with genomic instability are all involved in the development of resistance to monotherapy. Therefore, drug combination studies have emerged as a promising strategy to eradicate cancer cells and overcome their resistance to classical therapies. High-throughput screening has been used to determine the effects of pairwise drug combinations in solid tumors such as breast and colorectal cancer [56]. As a result, many combinations have already shown intriguing efficacy. Here are some examples of drug combinations in four of the most common solid tumors.

Breast cancer

Breast cancer is the leading cause of cancer death in women worldwide. In 2018, nearly 2.1 million new cases were identified, resulting in more than 625,000 deaths. Therefore, the treatment of breast cancer is a major challenge for the pharmaceutical industry and public research [57]. In addition, one of the major problems is the therapeutic resistance that often occurs in this cancer. This resistance can be characterized by breast cancer recurrence, which is the main cause of breast cancer-related deaths. Drug combinations have been largely assessed in breast cancer in order to obtain synergistic effects in killing tumor cells. A first example is the combination of doxorubicin (DOX) and glycyrrhetinic acid in a molar ratio of 1:20. This combination showed a synergistic effect against MCF-7 breast cancer cell lines, enhancing cytotoxicity and apoptosis and increasing intracellular DOX accumulation [58]. Another combination is that of paclitaxel (PTX) and verapamil (VERA) on adriamycin-resistant breast cancer cells (MCF-7/ADR). The IC50 values were significantly lower when both compounds were combined compared with PTX alone, as VERA enhanced the efficacy and sensitivity of PTX on these resistant cancer cells. Overall, the inhibition of cell proliferation appeared to be mediated by cell cycle arrest and apoptosis [59].

Colorectal cancer

Colorectal cancer (CRC) is the third most common cancer worldwide, accounting for over 1.85 million cases and 850,000 deaths annually. It is a common metastatic cancer, with 20% of cases diagnosed as metastatic and 25% of initially localized tumors eventually progressing to a metastatic stage. In addition, in the US, 40% of cases relapse after initial treatment of localized tumors, which is a significant challenge. In this context, the combination of pharmaceutical agents holds potential efficacy in combating resistance mechanisms and relapse [60].

For example, Limagne et al. demonstrated the efficacy of a combination of oxaliplatin and trifluridine/tipiracil (FTD/TPI) on the microsatellite-stable colorectal cancer cell line CT26. The importance of this combination became apparent during the transition from in vitro to in vivo experiments. Indeed, the addition of oxaliplatin appears to be essential for the induction of immunogenic cell death (ICD) in vivo [61]. In a separate study, Guo et al. showed that GW4064, an agonist of the farnesoid X receptor (FXR), enhances the chemosensitivity of CRC cells to oxaliplatin by inducing pyroptosis and apoptosis in vitro and by reducing tumor cell growth in vivo [62]. 

Prostate cancer

Prostate cancer was the second most commonly diagnosed cancer worldwide in 2020, with more than 1.4 million new cases and approximately 375,000 deaths, making it the fifth leading cause of death in men [63]. In addition, numerous resistance phenomena are observed, such as metastatic castration-resistant prostate cancer. Typically, the initial therapeutic approach involves androgen deprivation therapy to inhibit tumor cell progression. However, observed resistance phenomena undermine the efficacy of this initial form of treatment. The use of drug combinations is a promising way to combat such resistance. This was demonstrated in the study by Fizazi et al., which focused on the combination of abiraterone acetate and prednisone given alongside androgen deprivation therapy. This combination significantly increased overall survival and radiographic progression-free survival, making it a viable solution for men with metastatic castration-sensitive prostate cancer [64].

Lung cancer

In 2020, lung cancer alone accounted for 11.4% of all cancer cases worldwide, with more than 2.2 million cases and nearly 1.8 million deaths [65]. There are two main types of lung cancer, with non-small cell lung cancer (NSCLC) accounting for 85% of cases. The 5-year survival rate is only 16–18%, partly due to the asymptomatic nature of the early stages, which often delays the diagnosis [66]. Many efforts have been made to improve the prognosis, including combination drug therapies.

For example, in NSCLC driven by a BRAF gene mutation (essential in regulating cell growth, proliferation, and survival), the combination of dabrafenib and trametinib demonstrated an overall response rate (ORR) of 67%, compared with an ORR of 33% for dabrafenib monotherapy. Here the dual therapy improves the ORR by a factor of two [66]. Another in vitro study showed promising results with the combination of trametinib and rapamycin, as shown in the work of Sun et al. Indeed, on rapamycin-resistant NSCLC cells, the addition of trametinib significantly enhanced the inhibitory activity of rapamycin, showing a synergistic effect. This combination arrested the cell cycle and induced apoptosis by regulating cyclin-dependent kinases 2/4 (CDK2/4). Suppression of the AKT, ERK, mTOR, and 4EBP1 pathways was also observed. These encouraging results were successfully translated in vivo using a xenograft mouse model [67]. 

As a perspective, the use of appropriate drug delivery systems for the release of multiple drugs is a relevant strategy to combat cancer resistance to conventional therapy. The vectorization and release of multiple drugs at the desired site remains a major challenge, and many options are emerging from the efforts of laboratories worldwide. However, there are several Gordian knots in this strategy that need to be considered. For example, the difference in the hydrophilicity of drugs can complicate the design of nanocarriers, which must be able to transport two entities with different physical and chemical properties. This is illustrated in a study by Hu et al., who co-encapsulated hydrophilic doxorubicin hydrochloride with hydrophobic paclitaxel in a nanocarrier consisting of a hydrophilic core and a hydrophobic shell based on poly(vinyl alcohol) (PVA) and iron oxide. This nanocarrier, when exposed to an external stimulus (magnetic field), allows a 45–75% release over 10 h, depending on the molecular weight of the PVA [68]. 

### 3.4. Dual Inhibitors

Given the ubiquity of resistance and the significant side effects of current therapies, dual inhibitors (or multi-target inhibitors) are emerging as promising candidates to improve therapies and patient quality of life. In contrast to combination therapy, which can present challenges such as variable bioavailability, different metabolisms, or negative interactions between molecules, the dual-inhibitor strategy has the advantage of ensuring the presence of drugs at the disease site while achieving multi-target effects. For example, Zhang et al. described the synthesis of a hybrid compound, curcumin-BTP, which exhibited robust and selective anticancer activity against MCF-7 and doxorubicin-resistant MCF-7/DOX breast cancer cell lines (with IC50 values of 0.52 μM and 0.40 μM, respectively), while having a minimal effect on normal breast epithelial cells MCF-10A (Figure 14) [69]. This study also showed that curcumin-BTP inhibited STAT3-mediated P-glycoprotein expression in MCF-7/DOX cells and increased intracellular ROS production and accumulation. Overall, the decrease in IC50 appeared to be caused by cell cycle arrest and apoptosis. These findings are consistent with the original hypothesis to combine STAT3 inhibition with an “oxidation” therapeutic approach using dual inhibitors. 

As the concept of dual inhibitors appears to be relevant in the fight against cancer, scientists have begun to develop strategies for designing inhibitors capable of targeting different biological entities, each with different interactions and conformations. There are several strategies for designing such dual inhibitors.

Roskoski developed a classification of small-molecule kinase inhibitors, outlining seven different types. Some bind to the ATP pocket of active kinase conformations (type I), while others can bind to the same pocket but in an inactive conformation. In the latter case, when a DFG motif (Asp-Phe-Gly) commonly found in kinase scaffolds is engaged with an inhibitor while facing outward, the inhibitor is classified as type II [70]. Numerous compounds such as imatinib and sorafenib have been confirmed as type II inhibitors by crystallography. Through this crystallographic analysis, a model was constructed by Tan et al. that shows the importance of specific pharmacophores for this type of inhibitor. By following this general model, designed inhibitors stand a chance of having inhibitory activity against multiple targets, although there is a potential risk in terms of kinase selectivity [71]. Another strategy is to synthesize hybrids of multiple inhibitors targeting different biological entities. Identifying which part of each inhibitor influences its inhibitory activity is crucial for combining different pharmacophores within a single molecule to create a multi-target inhibitor. This was achieved by Tanaka et al. using two inhibitors of anti-apoptotic Mcl-1 and Bcl-xL proteins. The individual inhibitors had IC50 values of 0.54 and 0.15 µM, whereas the hybrid had IC50 values of 0.61 µM against Mcl-1 and 4.4 nM against Bcl-xL (Figure 15), demonstrating the effectiveness of this design strategy [72].

A third strategy, the opposite of the second, focuses on the similarities between two structurally close proteins. In fact, even if the peptide sequences are slightly different, it may be possible to design a molecule capable of inhibiting both proteins. For example, Bruncko et al., who observed structural similarities between the Bcl-2 and Bcl-xL proteins, used molecular docking to identify a pharmacophore that showed inhibitory activity against both proteins at sub-nanomolar concentrations [73].

Furthermore, by searching for a promising protein combination to target, it becomes possible to screen specific inhibitors of one protein while evaluating them against another. Once a hit compound has been identified, pharmacomodulations can be performed to enhance activity against both targets. This is exactly what Apsel et al. did for the combined inhibition of tyrosine kinases and PI3-K family kinases. Indeed, the potential efficacy of simultaneous inhibition of these two types of kinases has already been demonstrated [74]. The initial identification of two structures with activities at micromolar concentrations for pharmacomodulation led to the creation of two new molecules with activities in the nanomolar range [75]. 

Another option for identifying dual inhibitors is high-throughput screening techniques using computer modeling and large libraries of inhibitor molecules targeting, for example, proteases or kinases. These libraries may be commercially available and provide easy access to inhibitor structures as well as relevant biological targets. For example, multi-target virtual ligand screening can classify compounds from these libraries based on their ability to inhibit one or more biological targets, using docking scores or detailed interactions [76].

The latest approach presented here, known as the similarity ensemble approach (SEA), helps to identify off-target effects of multi-target molecules by comparing similarities between cytotoxic agents. The field of computational chemistry, commonly referred to as computer-aided drug design, significantly supports and accelerates the discovery of new dual inhibitors, although laboratory testing is still required [77].

## 4. Conclusions

For decades, many patients have experienced resistance to their treatments. Some even experience relapses several weeks or years after the end of their treatment, often with increased aggressiveness of symptoms. What is more, it is well known that cancer treatments are associated with significant side effects that dramatically reduce patients’ quality of life. For this reason, the global cancer research community is striving to fully exploit the possibilities that lie ahead. Over the years, promising results have been observed, particularly with drug combinations, dual inhibitors, and targeted therapies. Breast cancer, for example, is increasingly well controlled, with many options available. Lung cancer, on the other hand, still has a rather bleak prognosis. This highlights the many challenges that still need to be overcome as resistance phenomena are becoming increasingly adaptive. 

## Figures and Tables

**Figure 1 cancers-16-00461-f001:**
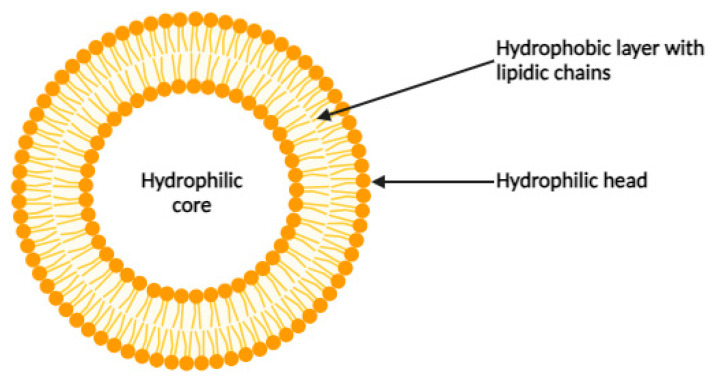
Schematic representation of a liposome.

**Figure 2 cancers-16-00461-f002:**
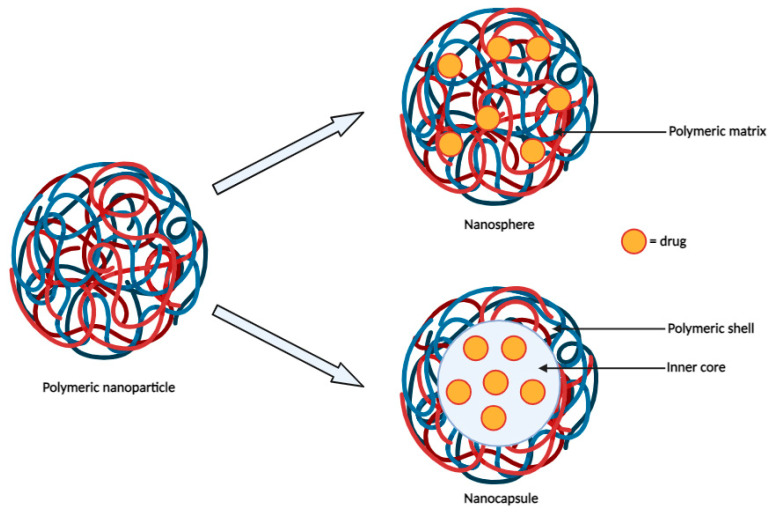
Schematic representation of polymeric nanoparticles.

**Figure 3 cancers-16-00461-f003:**
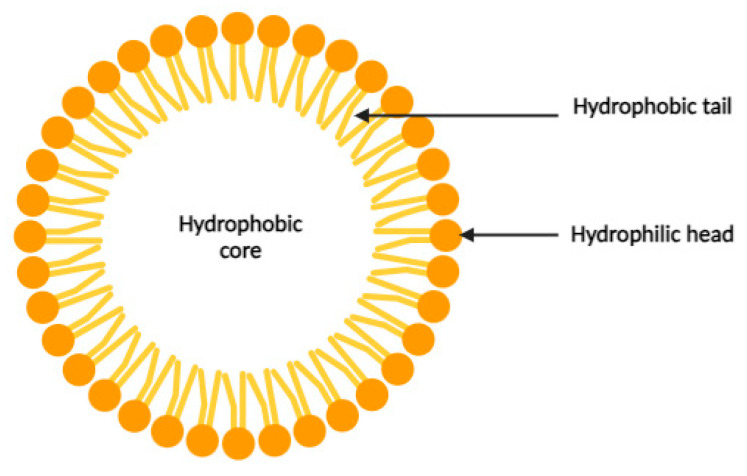
Schematic representation of a micelle.

**Figure 4 cancers-16-00461-f004:**
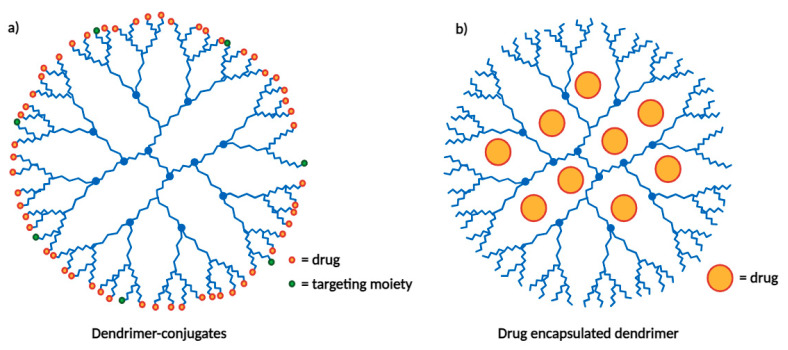
Potential applications of dendrimers. (**a**) Dendrimer-drug conjugates linked to targeting moieties. (**b**) Drug encapsulation within the dendritic interior.

**Figure 5 cancers-16-00461-f005:**
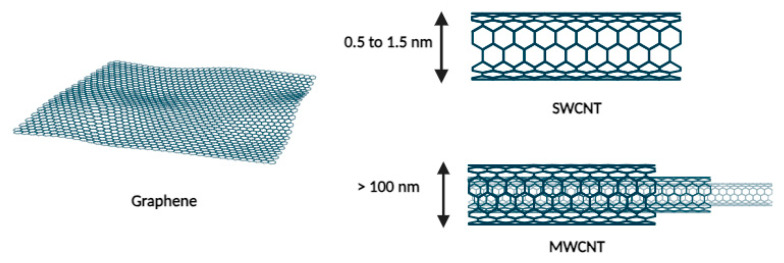
Representation of carbon nanotubes.

**Figure 6 cancers-16-00461-f006:**
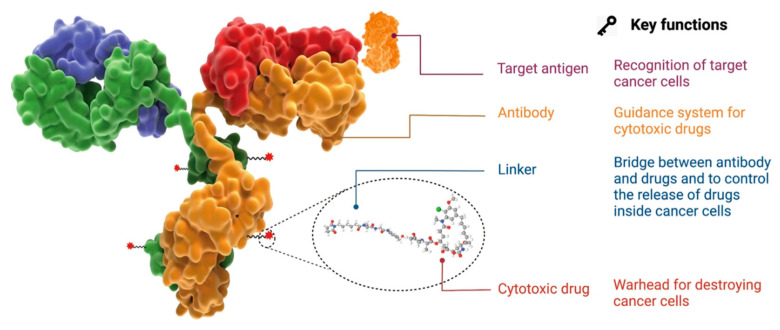
The structure and properties of an ADC drug. From Fu et al. [25].

**Figure 7 cancers-16-00461-f007:**
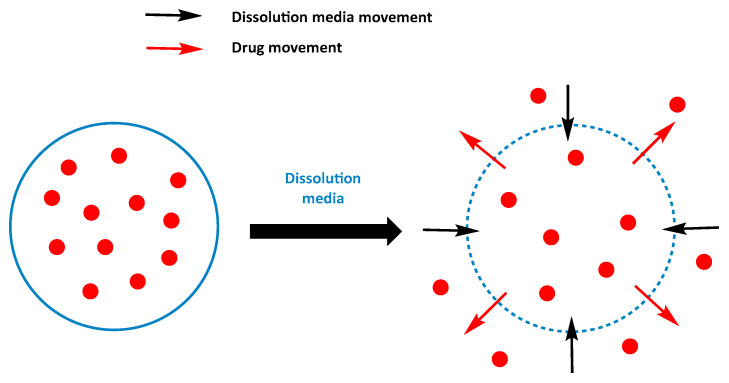
Mechanism of drug release from dissolution and diffusion-controlled drug delivery systems. From Qureshi et al. [35].

**Figure 8 cancers-16-00461-f008:**
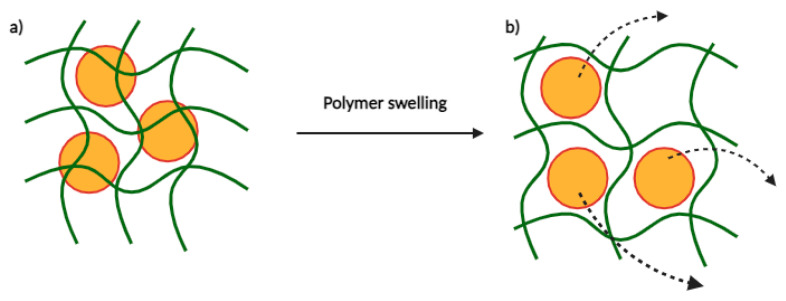
Drug diffusion by polymer swelling. (**a**) Drug encapsulated in polymer. (**b**) Drug release after polymer swelling.

**Figure 9 cancers-16-00461-f009:**
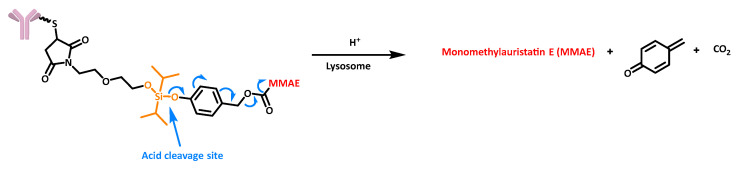
Structure and release mechanism of an ADC containing an acid-cleavable silyl ether-based linker.

**Figure 10 cancers-16-00461-f010:**
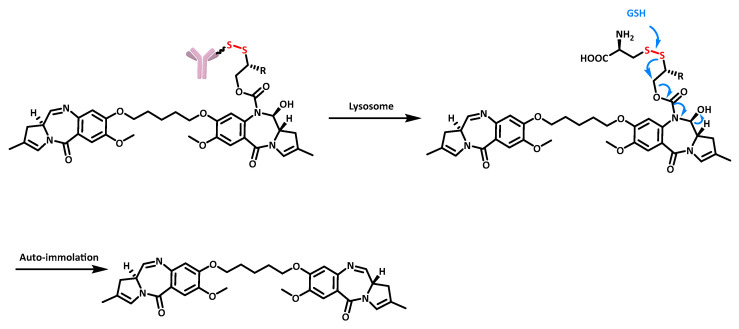
Structure and release mechanism of an ADC containing a disulfide–carbamate linker.

**Figure 11 cancers-16-00461-f011:**
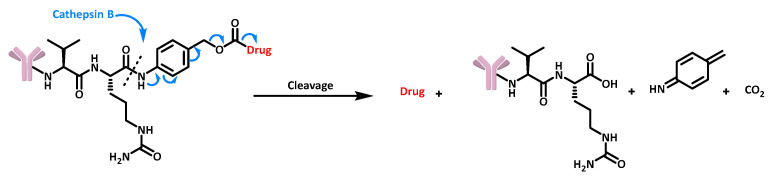
Structure and release mechanism of an ADC containing a cathepsin B cleavable linker.

**Figure 12 cancers-16-00461-f012:**
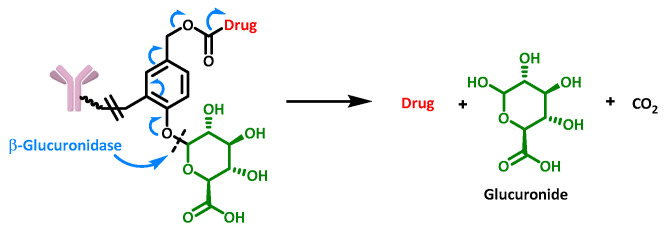
Structure and release mechanism of an ADC containing a β-glucuronidase cleavable linker.

**Figure 13 cancers-16-00461-f013:**
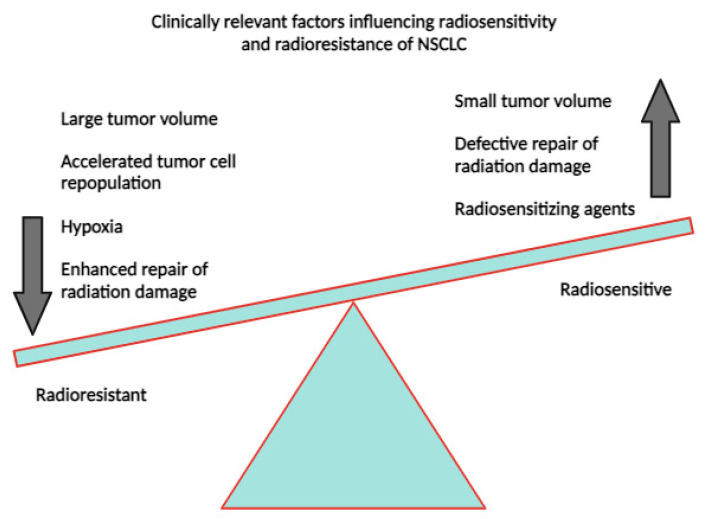
Known and putative factors influencing radiosensitivity and radioresistance in lung cancer. From Willers et al. [45].

**Figure 14 cancers-16-00461-f014:**
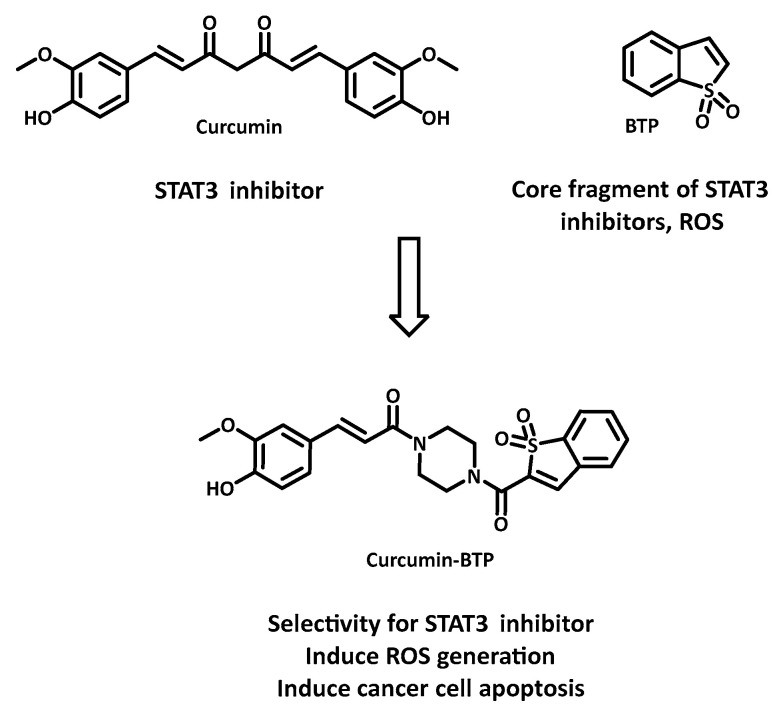
Design of the dual breast cancer inhibitor curcumin-BTP.

**Figure 15 cancers-16-00461-f015:**
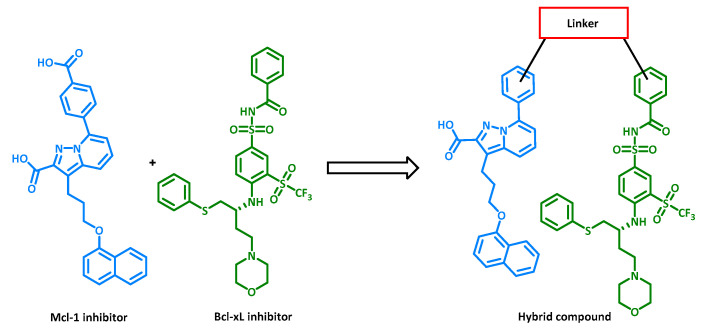
Hybridization strategy used to design Mcl-1/Bcl-xL dual inhibitors.

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
