# Peer review of "Targeted Therapeutic Strategies for the Treatment of Cancer"

_cancers, 2024, doi:10.3390/cancers16020461_

Round 1
Reviewer 1 Report
Comments and Suggestions for Authors
The work is a summary of a new therapeutic strategies to counteract therapy resistance in cancers. Authors presents various strategies to achieve this objective. They focus on different vectorization platforms capable of releasing drugs in cancer cells and also multitarget therapies using drug combinations and dual anticancer agents.
The work is interesting, after reading it there are only minor editorial comments:
1. The number of the cited work placed in square brackets should be written with a space and not placed directly after the word, e.g. chemotherapy[4] - it should be chemotherapy [4]
2. The terms in vitro and in vivo should be written in italics
3. On page 15, line 486, the authors quote the publication by Rosloski et al. [60] while in the References the publication with this number begins with the surname Roskoski
4. Personally, I am in the opinion that the article lacked a short subchapter describing current research directions regarding the issue presented in the work. What I mean here is Aptamer Liposome Conjugation, Oligonucleotide Liposome Conjugation, Antibody-siRNA Conjugation (ARC) & Fragment-Drug Conjugation (FDC), Antibody-Photosensitizer Conjugation (APC), Virus-like Drug Conjugation (VDC), Antibody-PNA or ASOs, Conjugation, Aptamer−Drug Conjugates.
However, the authors emphasize that in the article they present various existing options for the simultaneous administration of distinct drugs e.g. platforms capable of releasing drugs in cancer cells as well as multitarget therapies using drug combinations and dual anticancer agents. Therefore, the article in its current form may also be recommended for publication in Cancers.
Author Response
We are very pleased with the positive assessment by reviewer 1.
Point to point reply:
-Point 1-3: We have made the minor corrections requested by reviewer 1.
-Point 4: Personally, I am in the opinion that the article lacked a short subchapter describing current research directions regarding the issue presented in the work. What I mean here is Aptamer Liposome Conjugation, Oligonucleotide Liposome Conjugation, Antibody-siRNA Conjugation (ARC) & Fragment-Drug Conjugation (FDC), Antibody-Photosensitizer Conjugation (APC), Virus-like Drug Conjugation (VDC), Antibody-PNA or ASOs, Conjugation, Aptamer−Drug Conjugates.
We have followed the valuable suggestion of reviewer1, but to be consistent with the manuscript, we have now included a brief description of aptamer and oligonucleotide-liposome conjugates in section 2.1.1 (page 3, lines 74 to 80) and other bioactive conjugates in section 2.1.6 (page 7, lines 199 to 218).
Reviewer 2 Report
Comments and Suggestions for Authors
Authors reviewed the targeted therapeutic strategies for the treatment of cancer that has gained significance especially in therapy resistance and delivery. This review presents various strategies to achieve this objective, exemplifying diverse methods with concrete applications. The review scopes vastly the recent discoveries and applications in the field.
Some minor comments:
1. Rows 39-41: "Other factors are associated 39 with the inactivation of anticancer drugs through adaptive mechanisms, such as the inhibition of cell death or even multidrug resistance mediated by higher drug efflux [2]". This sentence is repetition of the previous.
2. Section 3.3 was not sufficiently cited. I propose the following references to strengthen the citating impact:
a. "Drug Combination in Cancer Treatment—From Cocktails to Conjugated Combinations" Cancers 2021, 13(4), 669; https://doi.org/10.3390/cancers13040669
b. "Chapter 2 - Overcoming the challenges of drug resistance through combination drug delivery approach" Academic Press, 2022, Pages 31-46, https://doi.org/10.1016/B978-0-323-85873-1.00003-4.
Author Response
We thank the reviewer 2 for the critical reading of our manuscript.
Changes were made according to reviewer’s comments. We have now introduced the requested references in section 3.3.